# Unleashing the Potential of Regularization Strategies in Learning with Noisy Labels

## Abstract

In recent years, research on learning with noisy labels has focused on devising novel algorithms that can achieve robustness to noisy training labels while generalizing to clean data. These algorithms often incorporate sophisticated techniques, such as noise modeling, label correction, and co-training. In this study, we demonstrate that a simple baseline using cross-entropy loss, combined with widely used regularization strategies like learning rate decay, model weights average, and data augmentations, can outperform state-of-the-art methods. Our findings suggest that employing a combination of regularization strategies can be more effective than intricate algorithms in tackling the challenges of learning with noisy labels. While some of these regularization strategies have been utilized in previous noisy label learning research, their full potential has not been thoroughly explored. Our results encourage a reevaluation of benchmarks for learning with noisy labels and prompt reconsideration of the role of specialized learning algorithms designed for training with noisy labels.

## 1 Introduction

Deep neural networks (DNNs) have become an essential tool for supervised learning tasks, and achieved remarkable progress in a wide range of pattern recognition tasks He et al. (2016); Girshick et al. (2014); Ronneberger et al. (2015); Devlin et al. (2018). These models tend to be trained on large curated datasets with high-quality annotations. Unfortunately, in many real-world applications such datasets are not available. However, datasets with lower-quality annotations, obtained from search engines or web crawlers Song et al. (2019); Xiao et al. (2015), may be available. When trained on these datasets, DNNs tend to overfit to the noisy labels, a consequence of overparameterization. This overfitting subsequently hampers DNNs' ability to generalize effectively, undermining their overall performance.

Recently, significant advances Reed et al. (2015); Goldberger & Ben-Reuven (2017); Malach & Shalev-Shwartz (2017); Han et al. (2018); Xu et al. (2019); Xia et al. (2020b); Yao et al. (2021) have been made to tackle the label noise problem using the ideas of modeling noise-label transition matrix Patrini et al. (2017); Xia et al. (2020a;b); Yao et al. (2020); Liu et al. (2022b), label correction/refurbishment Reed et al. (2014); Song et al. (2019); Zheng et al. (2020); Chen et al. (2021b); Liu et al. (2022a), co-training Jiang et al. (2018); Han et al. (2018); Yu et al. (2019), etc. These works often proceed with different assumptions regarding noise distributions or present sophisticated techniques to refurbish labels. Some other works are devoted to designing loss functions Zhang & Sabuncu (2018); Wang et al. (2019); Ma et al. (2020) and regularizations Liu et al. (2022b); Zhou et al. (2021) to prevent overfitting to label noise, and show better generalization ability. It is worth noting that almost all of these methods have adopted some *de facto* techniques for training modern neural networks e.g. learning rate decay, data augmentations, weight decay, etc. Focusing on tasks with noisy labels, a question that is naturally raised is how effective *de facto* regularization strategies are in achieving robustness to label noise?

In this paper, we propose an extremely simple baseline that suggests *de facto* regularization techniques can be more powerful for noisy classification tasks than the current crop of complicated algorithms dedicated to label noise. Our baseline consists of a set of regularization strategies. First, we use a sharp learning rate decay, where an initially large learning rate is used to avoid fitting noisy data, followed by a sharp decay to learn more complex patterns. Second, we adopt data augmentations

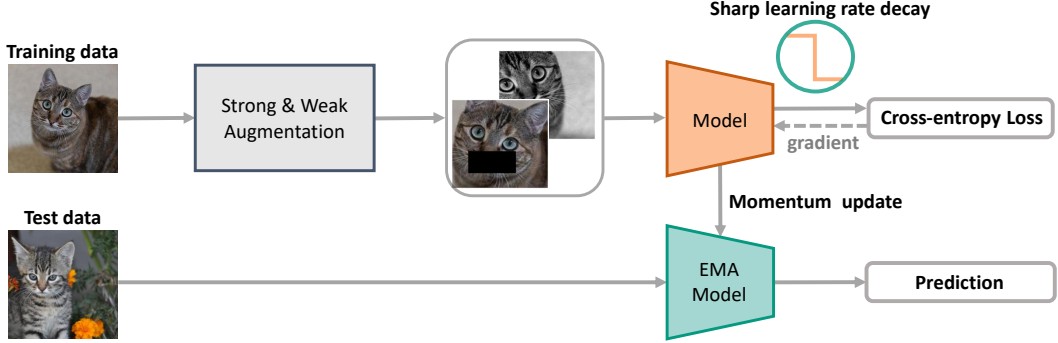

Figure 1: Overview of the proposed regularized cross-entropy (RegCE) method. The model is trained with a cross-entropy loss and some *de facto* regularization strategies: weak and strong augmentations, and multi-step learning rate decay. During training, the augmented data are fed into the model, the learning rate is large and scheduled to sharply decrease when training loss stops improving. The EMA model is updated as the exponential moving average of the model weights and adopted offline for testing.

with stronger transformation policies, including transformations such as cutout DeVries & Taylor (2017a), grayscale, color jitter, and Gaussian blur, etc. Third, we average model weights to smooth out variations during training.

Specifically, employing the three aforementioned regularization strategies, we train a DNN model using a standard cross-entropy loss with stochastic gradient descent (SGD) on noisily labeled datasets. We refer to this baseline method as Regularized Cross-Entropy (RegCE). Figure 1 provides an overview of the proposed RegCE. RegCE offers a contrast to current state-of-the-art methods Li et al. (2020); Liu et al. (2020; 2022b); Bai et al. (2021), which often involve more complex mechanisms. It is worth noting that current prevailing methods frequently adopt semi-supervised learning techniques to enhance performance. Therefore, we investigate the potential benefits of incorporating semi-supervised learning techniques into our method. In this exploration, we classify confident samples as labeled data and the remaining samples as unlabeled data, aligning with the practices employed by current state-of-the-art models. In summary, our main contributions are as follows:

- We propose a surprisingly simple baseline for learning with noisy labels, which achieves the state-of-the-art. This baseline suggests that many recent noisy label robust algorithms are *no better* than simply adopting multiple *de facto* regularization strategies together.

- We show that the simple baseline is a good base model for which the performance can be further improved by semi-supervised learning techniques.

- We support our findings with extensive empirical results on a variety of datasets with synthetic and real-world label noise.

## 2 RELATED WORK

**Learning with Noisy Labels.**    Recent advances in training with noisy label use varying strategies of noise-modeling and noise-modeling free approaches.

Model-based methods Patrini et al. (2017); Xia et al. (2020a;b); Yao et al. (2020); Liu et al. (2022b) strive to establish the relationships between noisy and clean labels, based on the assumption that the noisy label originates from a conditional probability distribution over the true labels. As a result, the primary goal of these methods is to estimate the underlying noise transition probabilities. Ref. Goldberger & Ben-Reuven (2017) employed a noise adaptation layer on top of a classification model to learn the transition probabilities. T-revision Xia et al. (2019) introduced fine-tuned slack variables to estimate the noise transition matrix without anchor points. Additionally, Ref. Liu et al. (2022b) proposed modeling label noise using a sparse over-parameterized term. These methods often assume certain characteristics about the noisy label distribution which may not hold for real-world.

In contrast to directly modeling the noisy labels, noise-modeling free methods Li et al. (2020); Xia et al. (2020a); Liu et al. (2020) aim to leverage the memorization effect of deep models to mitigate the negative impact of the noisy labels. Co-teaching Han et al. (2018) employs two deep networks to train each other using small-loss instances in mini-batches. ELR Liu et al. (2020) proposes regularizing training using model outputs at the early stage of training. PES Bai et al. (2021) selects different early stopping strategies for various layers of the deep model. Moreover, PADDLES Huang et al. (2022) proposes early stop at different stages for phase and amplitude spectrums of features. Despite the effectiveness of these approaches, they are often equipped with complex training steps such as two-network training, or dedicated methodology design for training with label noise.

**Regularization Techniques for Training Modern Neural Networks.** When training deep neural networks, it is often important to control overfitting with the helps from different forms of regularization techniques. Regularization can be implicit and explicit. Explicit regularization techniques, such as dropout Srivastava et al. (2014) and weight decay Loshchilov & Hutter (2017), reduce the effective capacity of the model. When noise is present, learning rate as "the single most important hyper-parameter" Bengio (2012) is shown to be an effective regularizer Li et al. (2019). A large initial learning rate is able to help escape spurious local minima that do not generalize LeCun et al. (1990); Kleinberg et al. (2018) and avoid overfitting noisy data Li et al. (2019); You et al. (2019). As another explicit regularization, average network weights along the trajectory of training is shown to find flatter minima and lead to better generalization Izmailov et al. (2018); Tarvainen & Valpola (2017). In contrast, data augmentation Perez & Wang (2017), as an implicit regularization, improves generalization by increasing the diversity of training examples without modifying the model's effective capacity Hernández-García & König (2018). Focusing on training with label noise, regularization techniques mentioned above are all widely used in state-of-the-art methods Liu et al. (2020); Nguyen et al. (2019) as a default setting without exploiting their full potential.

## 3 PROPOSED METHOD

In the context of learning with noisy labels, the true distribution of training data is typically represented by $\mathcal{D} = \{(x, y) \,|\, x \in \mathcal{X}, y \in 1, \ldots, K\}$. Here, $\mathcal{X}$ denotes the sample space, and $1, \ldots, K$ represents the label space consisting of $K$ classes. However, due to label errors during data collection and dataset construction, the actual distribution of the label space is often unknown. Therefore, we have to rely on a noisy dataset $\tilde{\mathcal{D}} = \{(x, \tilde{y}) \,|\, x \in \mathcal{X}, \tilde{y} \in 1, \ldots, K\}$ with corrupted labels $\tilde{y}$ to train the model. Our goal is to develop an algorithm that can learn a robust deep classifier from these noisy data to accurately classify query samples.

In the following, we first elaborate on the regularization strategies we suggest to be incorporated with standard training with cross-entropy loss: sharp learning rate decay schedule, strong and weak augmentations, and model weight moving average. Then, based on this regularized cross-entropy model, we present a learning algorithm that learns with consistent examples and semi-supervised learning techniques.

### 3.1 SHARP LEARNING RATE DECAY

Neural networks, due to their impressive model capacities, may inadvertently memorize and overfit noisy labels while training. As demonstrated by You et al. (2019), a larger initial learning rate can effectively suppress this unwanted noise memorization in the input samples. Our empirical observations affirm that this regularization effect also extends to noise present in labels. As depicted in Figure 2 (b), when a large initial learning rate (set to $0.1$) is used, the loss of clean samples decreases while the loss associated with noisy samples remains high. This indicates that the noise is not fully absorbed when the learning rate is large.

However, to optimize the fitting of clean labels and the learning of complex task-specific patterns, a reduction in the learning rate is required. We investigated various learning rate decay schedules commonly utilized in neural network training: cosine annealing decay, step decay, and gradual decay. Our findings suggest that a gradual decrease in the learning rate could lead the network to converge to a non-generalizable, sharp local minimum, as demonstrated in Figure 2 (c). Interestingly, we found that a sudden, sharp decrease in the learning rate could potentially enable the model to escape these spurious local minima, as depicted in Figure 2 (c, d).

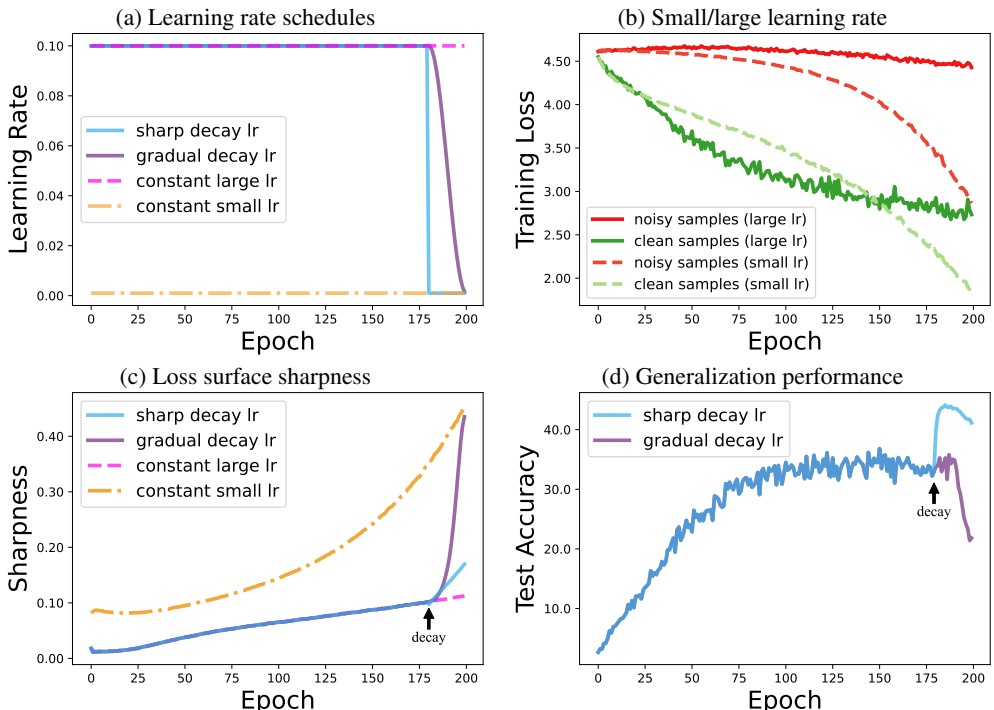

Figure 2: Comparison among different learning rate schedules. Figure (a) shows different learning schedules. The learning rate is set to be a constant with large (= 0.1) and small (= 0.001) values, or is decayed at the same iteration of training, while in two different ways: suddenly decayed or gradually decayed in a cosine curve. Figure (b) illustrates the average training loss of examples with clean and noisy labels with a constant learning rate. When the learning rate is large (= 0.1), cleanly labeled examples are fitted better than noisily labeled examples, while the learning rate is small (= 0.001), the noise is fitted faster, suggesting that a large initial learning rate can avoid overfitting to label noise. Figure (c) shows the sharpness of the loss surface under different learning rate schedules. Sharp learning rate decay may find minima with lower sharpness and better generalization. Figure (d) demonstrates the better generalization of sharply decaying the learning rate. Results from a ResNet-18 trained on CIFAR100 with 80% symmetric noise.

Hence, we recommend implementing a sharp learning rate decay strategy, where the learning rate is abruptly reduced to 1% of the original value once the training loss has stabilized. This is followed by another swift reduction of 1% after a few epochs of training. Our proposed learning rate scheduler, termed as the *sharp learning rate decay schedule*, exhibits robustness against label noise, as shown in Figure 2 (d).

## 3.2 DATA AUGMENTATIONS

Data augmentation, a broadly accepted approach for expanding datasets, is renowned for its ability to enhance model generalization and robustness Cubuk et al. (2019; 2020); Hendrycks et al. (2019). This technique entails applying diverse transformations to input data, thereby producing realistically viable variations. The gamut of these transformations spans from straightforward alterations like random cropping and flipping to more intricate operations such as cutout DeVries & Taylor (2017b) or image mixing Zhang et al. (2017). The gradations of these augmentations' complexity and intensity allow us to classify them into two categories: "weak" and "strong".

Weak augmentations are designed to introduce more subtle variations in the data, thereby ensuring a steady learning trajectory. Conversely, strong augmentations are tailored to incorporate more pronounced data variations, thus challenging the model to glean more robust and generalizable features. Nevertheless, when learning with noisy labels, weak augmentations often fall short in preventing model overfitting, while the distortions induced by strong augmentations could significantly alter image structures, complicating the learning process for the model.

Table 1: Comparison of test accuracy using different methods on CIFAR-10 and CIFAR-100 datasets with varying noise types and levels. The baseline results are taken from Yi et al. (2022). We use ResNet18 as the architecture, whereas all other methods in the comparison use PreAct ResNet18. The mean and standard deviation over 3 runs are reported. The best results are highlighted in bold.

| Dataset | Method | Symmetric | | | | Asymmetric |
| --- | --- | --- | --- | --- | --- | --- |
| | | 20% | 40% | 60% | 80% | 40% |
| CIFAR-10 | CE | 88.51±0.17 | 82.73±0.16 | 76.26±0.29 | 59.25±1.01 | 83.23±0.59 |
| | Forward | 88.87±0.21 | 83.28±0.37 | 75.15±0.73 | 58.58±1.05 | 82.93±0.74 |
| | GCE | 91.22±0.25 | 89.26±0.34 | 85.76±0.58 | 70.57±0.83 | 82.23±0.61 |
| | Co-teaching | 92.05±0.15 | 87.73±0.17 | 85.10±0.49 | 44.16±0.71 | 77.78±0.59 |
| | LIMIT | 89.63±0.42 | 85.39±0.63 | 78.05±0.85 | 58.71±0.83 | 83.56±0.70 |
| | SLN | 88.77±0.23 | 87.03±0.70 | 80.57±0.50 | 63.99±0.79 | 81.02±0.25 |
| | SL | 92.45±0.08 | 89.22±0.08 | 84.63±0.21 | 72.59±0.23 | 83.58±0.60 |
| | APL | 92.51±0.39 | 89.34±0.33 | 85.01±0.17 | 70.52±2.36 | 84.06±0.20 |
| | CTRR | 93.05±0.32 | 92.16±0.31 | 87.34±0.84 | **83.66±0.52** | 89.00±0.56 |
| | RegCE (ours) | **93.77±0.15** | **92.23±0.19** | **87.51±0.32** | 77.40±0.41 | **92.18±0.23** |
| CIFAR-100 | CE | 60.57±0.53 | 52.48±0.34 | 43.20±0.21 | 22.96±0.84 | 44.45±0.37 |
| | Forward | 58.72±0.54 | 50.10±0.84 | 39.35±0.82 | 17.15±1.81 | - |
| | GCE | 68.31±0.34 | 62.25±0.48 | 53.86±0.95 | 19.31±1.14 | 46.50±0.71 |
| | Co-teaching | 65.71±0.20 | 57.64±0.71 | 31.59±0.88 | 15.28±1.94 | - |
| | LIMIT | 58.02±1.93 | 49.71±1.81 | 37.05±1.39 | 20.01±0.11 | - |
| | SLN | 55.35±1.26 | 51.39±0.48 | 35.53±0.58 | 11.96±2.03 | - |
| | SL | 66.46±0.26 | 61.44±0.23 | 54.17±1.32 | 34.22±1.06 | 46.12±0.47 |
| | APL | 68.09±0.15 | 63.46±0.17 | 53.63±0.45 | 20.00±2.02 | 52.80±0.52 |
| | CTRR | 70.09±0.45 | 65.32±0.20 | 54.20±0.34 | 43.69±0.28 | 54.47±0.37 |
| | RegCE (ours) | **74.84±0.15** | **70.18±0.21** | **62.91±0.68** | **47.07±0.25** | **66.76±0.25** |

In light of these challenges, we propose an effective augmentation strategy that satisfies two primary conditions: (1) it enhances the model's generalization capabilities and mitigates the overfitting to noisy labels, and (2) it preserves the model's loss modeling and convergence properties without causing harmful impact. Our approach amalgamates the benefits of weak and strong augmentations to strike a balance between these requirements. Weak augmentations contribute to stability, while strong augmentations foster model robustness and resilience against noisy labels. This dual strategy is designed to harness the strengths of both augmentation types, ultimately boosting the model's performance when learning with noisy labels.

### 3.3 MODEL WEIGHTS AVERAGE

A unique challenge of learning with noisy labels often arises from the sequential fitting of the model to clean labeled samples, noisy labeled samples, and eventually, overfitting to all noisy samples Arpit et al. (2017); Liu et al. (2020). This progression frequently culminates in suboptimal generalization performance. To counteract this issue, we turned to the model weights averaging technique, derived from bootstrapping Efron & Tibshirani (1994). This powerful tool enhances model stability and boosts generalization performance Izmailov et al. (2018); Laine & Aila; Shi et al. (2018); Tarvainen & Valpola (2017), by aggregating model weights from different stages of training.

Notably, this regularization draws on the insight from Izmailov et al. (2018) that model weights averaging can facilitate the model's escape from local minima and guide it towards wider optima in the loss landscape. These wider optima represent more robust solutions and are associated with enhanced generalization performance. This feature is particularly beneficial when dealing with noisy labels, as it bolsters the model's robustness against label noise.

In this work, we adopted the exponential moving average (EMA) Hansun (2013) as our model weights averaging strategy. As illustrated in Figure 1, during training, we updated the online model using gradient back-propagation, while the offline EMA model was updated using the exponential moving average of the online model weights. The offline model is used as our final model for evaluation. This strategy, paying substantial attention to the early model weights primarily learned on clean labeled samples, helps preserve the model's capacity to classify clean samples accurately, a significant advantage when noisy labels are subsequently introduced afterward during training.

Table 2: Comparison of test accuracy using different methods under semi-supervised learning of confident examples on CIFAR-10 and CIFAR-100 datasets with varying noise types and levels. The baseline results are taken from Bai et al. (2021). All methods use ResNet18 as the base model. The mean and standard deviation over 3 runs are reported. The best results are highlighted in bold.

| Method | CIFAR-10 | | | CIFAR-100 | | |
|---|---|---|---|---|---|---|
| | 20% | 50% | 80% | 20% | 50% | 80% |
| CE | 86.5±0.6 | 80.6±0.2 | 63.7±0.8 | 57.9±0.4 | 47.3±0.2 | 22.3±1.2 |
| MixUp | 93.2±0.3 | 88.2±0.3 | 73.3±0.3 | 69.5±0.2 | 57.1±0.6 | 34.1±0.6 |
| DivideMix | 95.6±0.1 | 94.6±0.1 | 92.9±0.3 | 75.3±0.1 | 72.7±0.6 | 56.4±0.3 |
| ELR + | 94.9±0.2 | 93.6±0.1 | 90.4±0.2 | 75.5±0.2 | 71.0±0.2 | 50.4±0.8 |
| PES (semi) | 95.9±0.1 | 95.1±0.2 | 93.1±0.2 | 77.4±0.3 | 74.3±0.6 | 61.6±0.6 |
| RegCE + semi (ours) | **96.8±0.1** | **96.0±0.1** | **93.5±0.1** | **80.5±0.3** | **77.3±0.2** | **66.4±0.1** |

## 3.4 COMBINING WITH SEMI-SUPERVISED LEARNING

Currently, methods proposed for training with noisy labels often incorporate semi-supervised techniques to boost model performance Liu et al. (2022b; 2020); Li et al. (2020). Existing methods usually divide the dataset into confident samples and unconfident samples based on the model's predictions. Confident samples are used as labeled samples while unconfident samples are used as unlabeled samples. In order to successfully use these techniques, the reliability of a base model's prediction is vital. We show that our simple baseline RegCE is able to provide more accurate information to guide training in semi-supervised learning, which further boosts the model performance.

Following Ref. Bai et al. (2021), we obtain two views of the original inputs by data augmentations. We then use RegCE-trained model to obtain the predictions of the two views. We consider the samples in which the model's predictions are consistent between the two views and consistent to the label as labeled data and other inconsistent samples as unlabeled data. Once the data is divided into labeled and unlabeled subsets, we apply semi-supervised learning approach MixMatch Berthelot et al. (2019) to train the final model.

## 4 EXPERIMENT

### 4.1 DATASETS AND IMPLEMENTATION DETAILS

**Datasets:** We evaluate our method on two synthetic datasets with different noise types and levels, CIFAR-10 and CIFAR-100 Glorot & Bengio (2010), as well as three real-world datasets, CIFAR-N Wei et al. (2022), Animal-10N Song et al. (2019) and Clothing-1M Xiao et al. (2015). CIFAR-10 and CIFAR100 both contain 50k training images and 10k testing images, each with a size of 32×32 pixels. CIFAR-10 has 10 classes, while CIFAR-100 contains 100 classes. The original labels of these two datasets are clean. CIFAR-N consists of re-annotated versions of CIFAR-10 and CIFAR-100 by human annotators. Specifically, CIFAR-10N contains three submitted label sets (i.e., *Random 1, 2, 3*) which are further combined to have an *Aggregate* and a *Worst* label. CIFAR-100N contains a single human annotated label set named *Noisy Fine*. Animal-10N has 10 animal classes with 50k training images and 5k test images, each with a size of 64×64 pixels. Its estimated noise rate is around 8%. Clothing-1M has 1 million training images and 10k test images with 14 classes crawled from online shopping web sites.

**Synthetic Noise:** Following previous works Han et al. (2018); Liu et al. (2020); Xia et al. (2021); Patrini et al. (2017), we explore two different types of synthetic noise with different noise levels for both CIFAR-10 and CIFAR-100 datasets. For symmetric label noise in both datasets, each label has the same probability of being flipped to any class, and we randomly select a certain percentage of training data to have their labels flipped, with the range being {20%, 40%, 50%, 60%, 80%}. For asymmetric label noise in CIFAR-10, we follow the labeling rule proposed in Patrini et al. (2017), where we flip labels between TRUCK → AUTOMOBILE, BIRD → AIRPLANE, DEER → HORSE, and CAT ↔ DOG. We randomly choose 40% of the training data and flip their labels according to the asymmetric labeling rule. For asymmetric label noise in CIFAR-100, we also randomly select 40% of the training data and flip their labels to the next class in the label space.

Table 3: Comparison of test accuracy using different methods on real-world datasets Animal-10N and Clothing-1M. The baseline results are taken from Yi et al. (2022). All methods use ResNet18 as the base model. The mean and standard deviation over 3 runs are reported. The best results are highlighted in bold.

| Method | Animal-10N | Clothing-1M |
|---|---|---|
| CE | 83.18±0.15 | 70.88±0.45 |
| Forward | 83.67±0.31 | 71.23±0.39 |
| GCE | 84.42±0.39 | 71.34±0.12 |
| Co-teaching | 85.73±0.27 | 71.68±0.21 |
| SLN | 83.17±0.08 | 71.17±0.12 |
| SL | 83.92±0.28 | 72.03±0.13 |
| APL | 84.25±0.11 | 72.18±0.21 |
| CTRR | 86.71±0.15 | 72.71±0.19 |
| RegCE (ours) | **88.31±0.15** | **73.75±0.09** |

**Baseline Methods:** Semi-supervised learning (SSL) can significantly enhance performance. For fairness, we compare our proposed RegCE with and without SSL techniques separately.

For comparison without SSL, we primarily compare RegCE with several robust loss function methods: 1) Cross entropy loss. 2) Forward Patrini et al. (2017), which corrects loss values using an estimated noise transition matrix. 3) GCE Zhang & Sabuncu (2018), which takes advantage of both MAE loss and CE loss and designs a robust loss function. 4) Co-teaching Han et al. (2018), which maintains two networks and uses small-loss examples for updates. 5) LIMIT Harutyunyan et al. (2020), which introduces noise to gradients to avoid memorization. 6) SLN Chen et al. (2021a), which adds Gaussian noise to noisy labels to combat label noise. 7) SL Wang et al. (2019), which employs CE loss and a reverse cross entropy loss (RCE) as a robust loss function. 8) APL (NCE+RCE) Ma et al. (2020), which combines two mutually boosted robust loss functions for training. 9) CTRR Yi et al. (2022), which proposes a novel contrastive regularization function to address the memorization issue and achieves state-of-the-art.

For comparison with SSL, we utilize the previously mentioned RegCE+semi method, which incorporates MixMatch Berthelot et al. (2019), to compare it with other state-of-the-art LNL methods that are also combined with SSL techniques: 1) DivideMix Li et al. (2020), which first leverages SSL and a mixture model to effectively handle noisy labels. 2) CORES Cheng et al. (2021), which proposes a novel sample sieve framework to effectively identify and remove noisy instances during training. 3) ELR Liu et al. (2020), which utilizes the early-learning phenomenon to counteract the influence of the noisy labels on the gradient of the CE loss. 4) PES Bai et al. (2021), which presents a progressive early stopping method to better exploit the memorization effect of DNNs. 5) SOP Liu et al. (2022b), which models the label noise, and exploit implicit algorithmic regularizations to recover and separate the underlying corruptions.

**Implementation Details:** Our training parameters were set as follows: a batch size of 256, an initial learning rate of 0.1, and we implemented the SGD optimizer with a momentum of 0.9 and a weight decay of 5e-4. All experiments were conducted over a total of 200 epochs when not utilizing SSL techniques, and extended to 300 epochs when incorporating SSL techniques. Specifically, for experiments devoid of SSL, the learning rate was drastically reduced to 1% of the initial value when the training loss plateaued, and dropped another 1% after a few subsequent epochs of training. For experiments employing SSL, we initially trained for 100 epochs using RegCE to procure a robust initial model, which was then used to generate high-quality confident samples. These confident samples were then used for an additional 200 epochs of training, using the semi-supervised learning method MixMatch, as proposed in Berthelot et al. (2019). To stabilize the training process, we use the EMA across all experiments. Reported results are the mean and standard deviation computed over three independent runs. Further implementation details can be found in the Appendix.

## 4.2 CLASSIFICATION PERFORMANCE ANALYSIS

**Results on Synthetic Datasets:** As shown in Table 1, the proposed RegCE method was evaluated on the CIFAR-10 dataset under varying levels of label noise, achieving peak accuracy scores at

Table 4: Comparison of test accuracy using different methods on real-world dataset CIFAR-N. The baseline results are taken from the leaderboard in Wei et al. (2022). All methods use ResNet34 as the base model, except for SOP+ which uses PreAct ResNet18. The mean and standard deviation over 3 runs are reported. The best results are highlighted in bold.

| Method | CIFAR-10N | | | | | CIFAR-100N |
|---|---|---|---|---|---|---|
| | Random 1 | Random 2 | Random 3 | Aggregate | Worst | Noisy Fine |
| CE | 85.02±0.65 | 86.46±1.79 | 85.16±0.61 | 87.77±0.38 | 77.69±1.55 | 55.50±0.66 |
| Forward | 86.88±0.50 | 86.14±0.24 | 87.04±0.35 | 88.24±0.22 | 79.79±0.46 | 57.01±1.03 |
| T-revision | 88.33±0.32 | 87.71±1.02 | 87.79±0.67 | 88.52±0.17 | 80.48±1.20 | 51.55±0.31 |
| Co-Teaching | 90.33±0.13 | 90.30±0.17 | 90.15±0.18 | 91.20±0.13 | 83.83±0.13 | 60.37±0.27 |
| ELR+ | 94.43±0.41 | 94.20±0.24 | 94.34±0.22 | 94.83±0.10 | 91.09±1.60 | 66.72±0.07 |
| CORES* | 94.45±0.14 | 94.88±0.31 | 94.74±0.03 | 95.25±0.09 | 91.66±0.09 | 55.72±0.42 |
| DivideMix | 95.16±0.19 | 95.23±0.07 | 95.21±0.14 | 95.01±0.71 | 92.56±0.42 | 71.13±0.48 |
| PES (semi) | 95.06±0.15 | 95.19±0.23 | 95.22±0.13 | 94.66±0.18 | 92.68±0.22 | 70.36±0.33 |
| SOP+ | 95.28±0.13 | 95.31±0.10 | 95.39±0.11 | 95.61±0.13 | 93.24±0.21 | 67.81±0.23 |
| RegCE+semi (ours) | **96.55±0.03** | **96.51±0.08** | **96.69±0.07** | **96.41±0.20** | **94.80±0.07** | **74.28±0.22** |

noise levels of 20% and 40%. Even when the noise level increased to 60%, RegCE sustained its performance and continued to yield results comparable to those of CTRR at an extreme noise level of 80%. When applied to the CIFAR-100 dataset, RegCE consistently outperformed all other methods across all noise levels. For instance, at a noise level of 20%, it achieved an accuracy of 74.84%, markedly higher than CTRR's 70.09%. This performance trend was maintained up to a noise level of 80%, where RegCE achieved an accuracy of 47.07%, exceeding CTRR by more than three percentage points. The results validate the effectiveness of RegCE as a strong baseline in addressing label noise.

As detailed in Table 2, under the semi-supervised learning scenario, our method, RegCE+semi, consistently achieved the highest accuracy across all noise levels on both the CIFAR-10 and CIFAR-100 datasets. This result is noteworthy, considering the increased complexity of CIFAR-100 compared to CIFAR-10. For instance, at an extreme noise level of 80% on the CIFAR-100 set, RegCE+semi attained an accuracy of 66.4%, a score notably higher than the next best performing method, PES (semi), which reached 61.6%. These results suggest that our method can effectively utilize more accurate information to guide the training process in a semi-supervised learning environment.

**Results on Real-world Datasets:** Table 3 illustrates the superior performance of our proposed RegCE method on the Animal-10N and Clothing-1M datasets, compared to several baseline methods. By consistently outperforming all other methods, RegCE achieves the highest test accuracy of 88.31% and 73.75% on Animal-10N and Clothing-1M, respectively.

The efficacy of our RegCE+semi method is further highlighted in Table 4 on the CIFAR-N dataset, under a variety of label noise scenarios. Across all settings, RegCE+semi consistently secures the highest test accuracy. Notably, in the "Worst" scenario, characterized by the highest noise levels, our method excels above all others. Moreover, even under the demanding "Noisy Fine" scenario, RegCE+semi manages to maintain the highest accuracy, demonstrating its robustness and resilience against complex noise structures.

## 4.3 ABLATION STUDIES

In this section, we conduct ablation studies to analyze how each component affects the training with label noise performance. We study the three regularization strategies of our method: (a) we choose an initially large learning rate and then suddenly decay it to $1\%$ of the original value when the training loss does not improve; (b) we choose to apply both strong and weak augmentation together. We observe that solely using either weak or strong augmentation results in worse performance, and (c) we average the model's weight during training and use this averaged model for evaluation. We also evaluate the model's sensitivity to hyperparameters such as initial learning rate, weight moving average momentum, etc. More detailed analysis can be found in the Appendix.

Table 5 shows the results of our ablation studies on CIFAR-100 with $80\%$ of symmetric label noise. In general, each regularization strategy provides a performance boost, and the model is best performed with all three strategies used together. Figure 4 shows that solely using weak augmentation results in

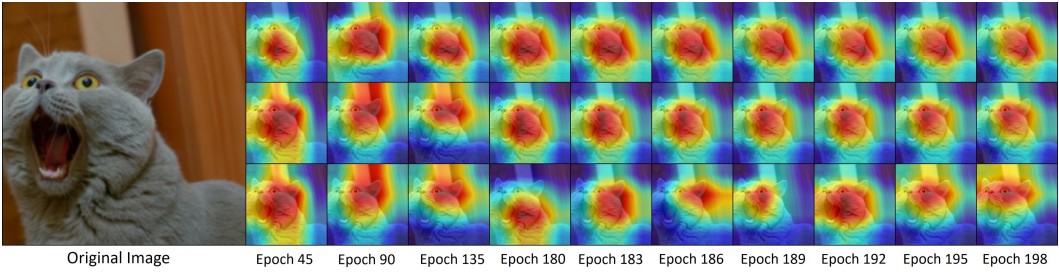

| Original Image | Epoch 45 | Epoch 90 | Epoch 135 | Epoch 180 | Epoch 183 | Epoch 186 | Epoch 189 | Epoch 192 | Epoch 195 | Epoch 198 |

Figure 3: Grad-CAM Selvaraju et al. (2017) results for different epochs of a ResNet-18 model trained on CIFAR-10. The **top** row presents the Grad-CAM of the model trained with true labels. The **middle** row shows results on 80% of symmetric noise with the proposed sharp learning rate decay. The **bottom** row represents the results on the same noisy dataset while with gradual learning rate decay.

overfitting to label noise while strong augmentation results in slow convergence, both result in worse performance.

Figure 3 illustrates the impact of learning rate decay schedules on models using Grad-CAM. Models robust to label noise consistently maintain Grad-CAM output during training when guided by the true label (middle row). Label noise may drive the model to absorb irrelevant background features. When label noise is minimal, the model accurately focuses on relevant features (top row). However, increasing label noise can misdirect the model's attention, lowering performance. A well-selected decay schedule can mitigate the effect of noisy labels, leading to a more task-focused model. Thus, a robust model can maintain consistent Grad-CAM output during training, emphasizing correct input regions despite label noise. Further research is needed to better understand the relationship between learning rate decay schedules, label noise, and model attention mechanisms for developing more robust models.

Table 5: Ablation study evaluating the influence of sharp learning decay, strong and weak augmentations, and model moving average. The mean accuracy computed over three noise realizations is reported.

| Method | | | | | | | | |
|---|---|---|---|---|---|---|---|---|
| LR | ✗ | ✓ | ✗ | ✗ | ✓ | ✓ | ✗ | ✓ |
| Aug | ✗ | ✗ | ✓ | ✗ | ✓ | ✗ | ✓ | ✓ |
| EMA | ✗ | ✗ | ✗ | ✓ | ✗ | ✓ | ✓ | ✓ |
| Test Acc. | 31.76 | 41.15 | 36.82 | 40.02 | 45.05 | 44.14 | 43.09 | **47.82** |

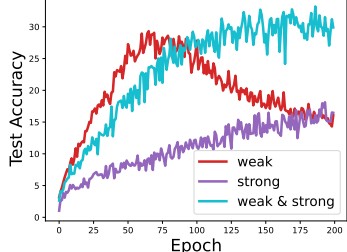

Figure 4: The performance comparison of different strategies of augmentations.

## 5 CONCLUSION

In this paper, we introduced RegCE, an extremely simple baseline method for learning with noisy labels that adopts several conventional regularization strategies. We demonstrated that this straightforward approach could match, and often outperform, the current state-of-the-art complex mechanisms developed specifically to tackle label noise. Our findings underline the surprising power of *de facto* regularization techniques such as multi-step learning rate decay, extensive data augmentation, and model weight averaging. These techniques, when combined, showed to be effective in enhancing the robustness of neural networks against label noise, which is often an overlooked aspect in the current literature. Furthermore, we explored the potential of incorporating semi-supervised learning techniques into our baseline method, which can further boost the performance. Our extensive empirical results on various datasets, both synthetic and real-world, further corroborate our claims. These findings motivate a rethinking of the prevalent complex approaches toward label noise and call for further exploration of the effectiveness of these simple regularization strategies for training with label noise. Future work could extend the application of our baseline method to other domains and tasks, as well as investigate other potential *de facto* regularization techniques that could further enhance the robustness of neural networks against label noise.

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

## A  ABLATION STUDIES

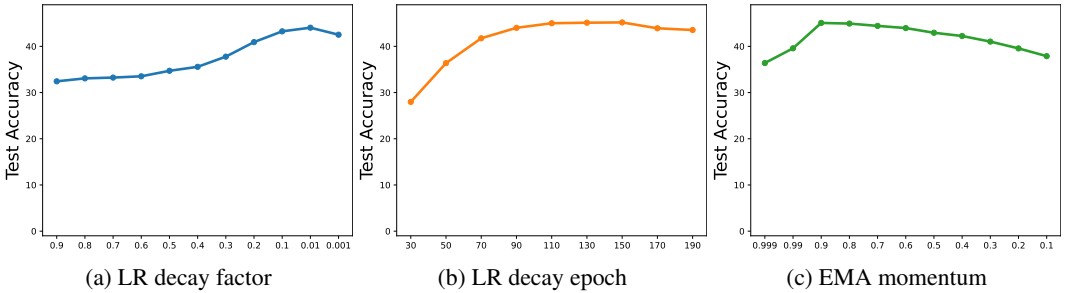

|     (a) LR decay factor     |     (b) LR decay epoch     |     (c) EMA momentum     |

Figure 5: The hyperparameter ablation studies of sharp learning rate decay and model moving average. Results from a ResNet-18 model trained on the CIFAR-100 dataset with 80% symmetric label noise.

Figure 5(a) presents the correlation between the learning rate (lr) decay rate and the corresponding test accuracy. It shows the fact that, after a period of training with a large learning rate, it becomes crucial to reduce the learning rate significantly. This strategy enables the model to better fit clean labels and discern complex patterns for the task, thereby boosting test accuracy. Figure 5(b) focuses on studying the impact of the sharp decay point. It shows that the model requires more training in the early stages to learn the data's features and patterns. Starting the decay too early may prevent the model from fully learning during this critical phase, resulting in decreased performance. Only by initiating the decay at the appropriate time can the model effectively leverage the training data and enhance performance. Figure 5(c) presents the momentum value of EMA model how to effect the test accuracy. Selecting an appropriate EMA momentum value is crucial. A low momentum value may cause the model to react too quickly to new model weights, potentially overfitting to recent trends and ignoring longer-term patterns. Conversely, a high momentum value may cause the model to react too slowly, potentially underfitting and not adequately capturing recent trends.

## B  TRAINING DETAILS

### B.1  DATA PREPROCESSING

Our preprocessing procedures for all datasets incorporate a combination of weak and strong augmentation techniques. Weak augmentations include random cropping and flipping, while strong augmentations comprise cutout, grayscale transformations, color jitter, and Gaussian blur. In addition to these, for experiments involving semi-supervised learning methodologies, we also utilize mixup, a vital element of the MixMatch technique. We perform two rounds of data augmentation on each image in a mini-batch of $N$ images, where $N$ is set to 128 in all experiments. This results in a larger batch size, that is, $2N$ images, serving as input to the model. These two rounds of augmentation serve different purposes: one for weak augmentations and the other for strong augmentations. As described in the main body of the paper, weak augmentations contribute to stability, while strong augmentations foster model robustness and resilience against label noise.

### B.2  TRAINING SETTINGS

For our research, we employed datasets of CIFAR-10, CIFAR-100, CIFAR-N, and Animal-10N, utilizing the entirety of the training data for each epoch. For these datasets, we initiated the process with a randomly initialized model and established an initial learning rate of 0.1. In the case of the Clothing-1M dataset, for a fair comparison consistent with the methodology used in Yi et al. (2022),

we randomly selected a balanced subset of 20.48K images from the noisy training data for every epoch, implemented an ImageNet pre-trained weights, and set the initial learning rate to 0.01.

In conducting experiments utilizing semi-supervised learning methodologies, we employ confident samples as labeled data, and unconfident samples as unlabeled data. It's important to note that the proportion of confident to unconfident samples can vary substantially depending on the noise levels in the settings. To tackle the potential issue of class imbalance, we also adopt a class balance strategy, following the approach outlined in Bai et al. (2021). The difference is that we use class-balanced samplers for the data loaders, and sample the data to the same amount of the original data. With regards to the subsequent training process, we adhere to the original MixMatch Berthelot et al. (2019) settings. Specifically, we set $K = 2, T = 0.5, \alpha = 0.75$, with $\lambda_u$ chosen from the set $\{5, 75, 150\}$.

