# OpenReview forum: "Unleashing the Potential of Regularization Strategies in Learning with Noisy Labels"
_ICLR.cc/2024/Conference — Submitted to ICLR 2024_

### Official Review · Reviewer_m1A7 · 2023-10-23

**Soundness:** 3 good
**Presentation:** 3 good
**Contribution:** 3 good
**Rating:** 6
**Confidence:** 5

**Summary:**

This work proposes to combine some widely-used de facto regularzation methods (often utilized in noisy label learning, self-supervised learing and semi-supervised learning) all together to achieve more robustness against noisy labels. Extensive experiments basically support  their claims.

**Strengths:**

- This works is generally well organized and well written. The assessment (most state-of-the-art methods are actually combination of many de facto techniques) is direct.
- This works carefully review related litereature in the area of noisy label learning.
- Extensive experiments on many experiments support their claims.
- The experiments shown in Figure 2 is good : )  The analysis of Figure 2's (b) is explicit which serves as an illustration of deep network memorization effect (proposed by Arpit's work).
- Authors are encouraged to fully open-source codes of this work, which certainly helps to broaden the influence of this work and stimulates the future studies from related noisy label learning community.

**Weaknesses:**

- Authors should reimplement experiments on Animal-10N and Clothing-1M of Table 3, with the same base model architecture to achieve a fair comparison.
- Personally, authors are encouraged to investigate how the base model architecture affect the training performance against the noisy labels, since different models (which are ofen over-parameterized) are of different abilities to encode data.
- The proposed method seems simple, but the computation complexity is not that simple (especially combined with MixMatch), and I reckon authors must know that limitation.

**Questions:**

See above.

---

### Official Review · Reviewer_Mfti · 2023-10-29

**Soundness:** 2 fair
**Presentation:** 2 fair
**Contribution:** 2 fair
**Rating:** 3
**Confidence:** 4

**Summary:**

The paper claims that a naive combination of three existing regularization methods is a strong base model for learning with label noise. The claim if properly verified can be impactful. However, in its current form, the claim is not justified in a solid way and there are many important details missing.

**Strengths:**

1. The paper claims to find that simply combining three regularization methods is a strong baseline for learning with label noise. The finding can be impactful if verified.
2. Good experiment results are shown against selected baselines.
3. The paper shows semi-supervised learning can further improve performance.

**Weaknesses:**

1. Writing can be improved.
2. More intuition and justification for the proposed method should be provided.
3. Experiment setup is not convincing.
4. For a paper that claims a simple and strong baseline, opensourcing code can be very helpful for the community.

**Questions:**

1. Section 3.2 says “we propose an effective augmentation strategy”, but the strategy is never introduced.
2. The citation format is weird. The references “author year” mix up with the main text.
3.  How gradual decay is performance? Sharp decay set lr to be 1% of the original. Does gradual decay decrease lr to be less than 1%? If yes, this could be unfair to gradual decay.
4. There are many regularization methods out there. Why a combination of the three works the best? is there any intuition? What if I combine more regularization methods?
5. "dropped another 1% after a few subsequent epochs of training". How many epochs? is the number of epochs the same for different datasets? any intuition on why do you need to drop another 1%?
6. Does the baseline methods use the same initial learning rate as the proposed method?

---

### Official Review · Reviewer_7dS9 · 2023-10-31

**Soundness:** 3 good
**Presentation:** 3 good
**Contribution:** 2 fair
**Rating:** 3
**Confidence:** 4

**Summary:**

The paper addresses the issue of noisy labels in datasets and posits that an amalgamation of established regularization techniques can surpass state-of-the-art (SOTA) methods. The authors employ a tailored learning rate decay schedule, various data augmentation techniques, and model weight moving averages, all in conjunction with standard cross-entropy loss. Their proposed framework demonstrates superior performance over existing SOTA methods on multiple datasets.

**Strengths:**

The paper conducts a comprehensive analysis of traditional regularization techniques specifically in the context of noisy labels. The authors demonstrate that their chosen combination of regularization methods consistently outperforms competing approaches.

**Weaknesses:**

1. The work's innovation is somewhat limited, as the proposed methodology essentially amalgamates pre-existing techniques and focuses primarily on hyperparameter optimization.

2. Both Table 1 and Table 2 indicate that the RegCE+MixMatch performs better than RegCE alone. This suggests that further improvements to other SOTA algorithms could be achieved through the regularization strategies utilized in RegCE. The authors should provide more comparative analysis so that readers can get intuitions about regularization methods for various SOTA algorithms.

**Questions:**

See the 'Weaknesses' section for points requiring clarification.

---

### Official Review · Reviewer_tMPC · 2023-11-03

**Soundness:** 2 fair
**Presentation:** 3 good
**Contribution:** 1 poor
**Rating:** 3
**Confidence:** 3

**Summary:**

The Deep Learning book [1], defines regularization as: “any modiﬁcation we make to a learning algorithm that is intended to reduce its generalization error but not its training error.”
Hence, all methods developed for label noise are by definition also regularization techniques. We can then categorize regularization techniques based on if they were originally developed for robustness to label noise (explicit) or not (implicit).

The goal is to question the development of explicit methods for label noise by showing the effectiveness of combining widely used implicit regularization strategies. The paper proposes a new method for label noise robustness by combining three implicit regularization techniques: i) learning rate scheduling, ii) data augmentation, and iii) using a network with exponential moving average (EMA) of weights for evaluation.

The learning rate schedule is chosen to be a standard step-wise learning rate that quickly reduces the learning rate from its initial high value to a small value late in training. The authors propose a multi-augmentation strategy, where each image is augmented with both a weak and a strong augmentation strategy, leading to a twice as large batch size. Lastly, the EMA network does not affect the training in any way, and is only used at evaluation.

The authors combine the method described above with the self-supervised learning method MixMatch. To separate training set into labeled and unlabeled data, the authors use the consistency-based approach developed by Bai et al. [2].

The authors conduct experiments on synthetic noise on the CIFAR datasets, as well as on natural noise on the CIFAR-N datasets, Clothing1M and Animal-10N.

**References.**

[1] Goodfellow I, Bengio Y, Courville A. Deep learning.

[2] Bai, Y., Yang, E., Han, B., Yang, Y., Li, J., Mao, Y., ... & Liu, T. Understanding and improving early stopping for learning with noisy labels.

**Strengths:**

The paper is well-organized with clear section headers making the high-level layout easy to follow. The authors report mean and standard deviation over three runs in most of their results, which provides an indication of the variability of the results. To better understand the different components of their method, the authors also perform an ablation study, which is great. The design choices made when developing the method become clear through experiments. The proposed method is evaluated on several noise rates, noise types, and datasets.

**Weaknesses:**

Most of my main criticisms can be found in the following sentence from the abstract:
“In this study, we demonstrate that a **simple** baseline using cross-entropy loss, combined with **widely used regularization strategies** like learning rate decay, model weights average, and data augmentations, can **outperform state-of-the-art methods**.”

**Goal and Method are Contradictory.**

The goal is to question the development of explicit methods for label noise by showing the effectiveness of widely used implicit regularization strategies. This is contradictory as e.g., in the appendix, the authors describe that it is not just the strength of the data augmentation strategy that is changed. Instead, a new multi-augmentation method is used, where two augmentation strategies are used at the same time, creating a twice as large batch. Therefore, the gradients for an original image get information from two augmentations at the same time, resulting in some averaged gradient instead. The introduction and effect of this seems poorly understood and investigated, and is far from the “widely used regularization strategy” of data augmentation. The only experiment done is to train using only weak, or only strong, or weak and strong augmentations at the same time (twice as large batch). The arguments for using weak and strong are not based on this gradient averaging effect. To investigate the influence of this unaccounted for effect, an experiment could be done where for each image, a weak or a strong augmentation is randomly chosen. I believe this also gets the augmentation approach closer to a “widely used regularization strategy”, as we only have one augmentation per image, leading to the original batch size.

**Limited novelty.**

From the definition in the introduction, it is clear that all regularization techniques are by definition likely to improve robustness to label noise. It is well-known that implicit regularization techniques improve robustness to label noise, e.g., via label smoothing [1], early stopping [2], learning rate scheduling [3], dropout [4], data augmentation [5], exponential moving average of weights [6], etc. There is no surprise either that many of these methods provide complementary robustness improvements, often shown in ablation studies (e.g., [6, 7]). To me then, the novelty of the paper then lies in the proposed changes to the implicit regularization techniques and in how well these can be made to work together, which leads to the next issue.

**Unfair and inconclusive comparisons.**

There are many regularization methods that explicitly tackles label noise, e.g., via robust loss functions, or via methods that identify and remove noisy examples, or methods that try to correct the noisy labels in the training set, etc.

When comparing different explicit label noise methods it is common to use the same and simple training setup in terms of implicit regularization techniques, e.g., only weight decay and crop and flip as data augmentation. The reason for having the same implicit regularization is that the robustness improvements of implicit techniques are often orthogonal to any explicit method. Furthermore, the reason for having simple implicit regularization is to have as few confounders as possible. Clearly, the goal isn’t to achieve the highest possible performance in this setup, but rather to have an as simple, fair and conclusive setup to compare and understand explicit methods in.

In this work the authors create an explicit regularization technique for label noise out of re-designed implicit regularization techniques. Therefore, the authors seem to think it is valid to compare their approach, with explicit methods that use the simple and most basic implicit regularization techniques (Tables 1 and 3), or with less strong implicit regularization techniques (Tables 2 and 4). However, leveling the playing field in terms of implicit regularization could significantly improve the performance of the explicit methods, making the comparison flawed.

Note that all baseline results in Tables 1, 2, 3, and 4, are all directly taken from several other papers, with potentially more differences in training/evaluation setup, making the comparisons inconclusive. In particular, the authors claim state-of-the-art performance. In the introduction, when referring to state-of-the-art methods, the authors cite: DivideMix, ELR/ELR+, SOP/SOP+, and PES. Comparisons with these baselines are done in Tables 2 and 4, where all baseline results are directly taken from other papers in each table. Therefore, these comparisons can be unreliable due to different training and evaluation setups. Indeed, it seems all the baselines in Tables 2 and 4 use Mixup with crop and flip (with the exception of SOP+ in Table 4), while RegCE+semi uses much stronger data augmentation strategies as it additionally uses cutout, gray scale transformation, color jitter, and Gaussian blur. Therefore, we cannot know if RegCE+semi is a better method or if it is only due to the unfair advantages in the training (e.g., stronger augmentations).

**Misleading simplicity claim.**

The word “simple” in the sentence from the abstract can refer to one of two things: RegCE or RegCE+semi. If simple refers to RegCE, then it is misleading to claim it can outperform the SOTA methods, as these were only compared with RegCE+semi not RegCE. Furthermore, I would argue that RegCE is a much more complex approach than, say, a simple replacement of CE to a robust loss function.

To me, it would be unreasonable/misleading to refer to RegCE+semi as simple as it is as complex (if not more) than the notoriously complex DivideMix (both have some criteria to divide examples, and use MixMatch, and RegCE also has some type of gradient averaging effect).


**Suggestions for improvement.**

I do believe it is important to understand the influence of implicit regularization techniques, but I see no value in only using implicit regularization strategies for robustness to label noise, as they can be as complex as any explicit method. It reminds me of the famous saying: "If the only tool you have is a hammer, it is tempting to treat everything as if it were a nail." Let’s use the hammer, but in combination with specialized tools for the task.

I would encourage the authors to study a single implicit regularization in detail to better understand its effect on noise robustness. Or better understand how the robustness of standard implicit regularization techniques (e.g., data augmentation, learning rate scheduling, and weight decay.) together help in a complementary way. I believe another valuable research contribution would be to investigate the influence of implicit regularization techniques on existing explicit techniques. That is, implement several baselines in the same training setup and vary the strength of the implicit regularization techniques to see how big of an improvement one can achieve and which methods benefit the most and why. This would be especially interesting on real-world noise benchmark datasets, e.g., WebVision and Clothing1M.



**References.**

[1] Wei J, Liu H, Liu T, Niu G, Sugiyama M, Liu Y. To smooth or not? when label smoothing meets noisy labels.

[2] Li M, Soltanolkotabi M, Oymak S. Gradient descent with early stopping is provably robust to label noise for overparameterized neural networks.

[3] Tanaka D, Ikami D, Yamasaki T, Aizawa K. Joint optimization framework for learning with noisy labels.

[4] Jindal I, Nokleby M, Chen X. Learning deep networks from noisy labels with dropout regularization.

[5] Nishi K, Ding Y, Rich A, Hollerer T. Augmentation strategies for learning with noisy labels.

[6] Liu S, Niles-Weed J, Razavian N, Fernandez-Granda C. Early-learning regularization prevents memorization of noisy labels.

[7] Li J, Socher R, Hoi SC. Dividemix: Learning with noisy labels as semi-supervised learning.

**Questions:**

* In the Appendix, the “data augmentation” process is described, where it is clarified that the data augmentation strategy is not only changed, but also the batch is made twice as big, one with weak and one with strong. You also mention that only weak or only strong does not work as well.
  * Isn’t it misleading to say that the method is a combination of “widely used regularization strategies”, then?
  * What is the influence of the gradient averaging effect on robustness?
* Could the authors clarify what makes the methods novel and interesting to the community? Why would a combination of only implicit regularization techniques into an explicit method for label noise be desired over an explicit method or a combination of explicit and implicit? For example, a very simple but interesting experiment would be to keep everything the same in RegCE/RegCE+semi, but replace the CE loss with the GCE loss and see if there are any performance improvements compared to RegCE/RegCE+semi.
* Not a single baseline was re-implemented and evaluated in your training setup, i.e., all results in the paper for baselines are taken from other papers. How does this affect the reliability and fairness of your comparisons and therefore your conclusions?
  * Do you think the non-standard change you make to the data loader for the experiments with the additional semi-supervised learning component, mentioned in the last paragraph in the appendix of the paper, is fair compared to baselines? How does this influence the results?
  * More generally, what are the training and evaluation differences between your results and each of the papers you have directly taken results from? For example, differences in strengths or methods used for implicit regularization, the size of the training set, differences in learning rate, etc.
	* It was mentioned the full training set was used for your results. Was this also done in all the papers the baseline results have been taken from?
	* Furthermore, if the full training set was used, how were the optimal hyper-parameters chosen? Test set? How were hyper-parameters selected for your method? One set of HPs per dataset or per noise rate etc? How does the HP search for your method differ compared to the baselines results that were taken from other papers?

* Could the authors clarify in what way the proposed methods are “simple”? Compared to what?

* Could the authors elaborate on the following sentence from the abstract: “Our results encourage a reevaluation of benchmarks for learning with noisy labels and prompt reconsideration of the role of specialized learning algorithms designed for training with noisy labels.”
  * Why and how should benchmarks be reevaluated? Why and how should the role of specialized learning algorithms be reconsidered?

---

### Meta-Review · Area_Chair_DpCV · 2023-12-04

**Metareview:**

The paper combines to propose a new regularization technique. The main issues raised by the reviewers are: 1) the technical novelty is limited, and 2) the proposed method is somewhat complex because it requires, e.g., carefully tuning a learning rate. Overall, the concerns raised by the reviewers remain unresolved.

**Justification For Why Not Higher Score:**

The overall score is quite low, and there are several concerns that need to be addressed.

**Justification For Why Not Lower Score:**

N/A

---

### Decision · Program_Chairs · 2024-01-16

Reject